# Evaluating The Search Phase of Neural Architecture Search

**Kaicheng Yu**[*]
Computer vision lab, EPFL
kaicheng.yu@epfl.ch

**Christian Sciuto**[*†]
Daskell
christian.sciuto@daskell.com

**Martin Jaggi**
Machine learning and optimization lab, EPFL
martin.jaggi@epfl.ch

**Claudiu Musat**
Swisscom Digital Lab
claudiu.musat@swisscom.com

**Mathieu Salzmann**
Computer vision lab, EPFL
mathieu.salzmann@epfl.ch

## Abstract

Neural Architecture Search (NAS) aims to facilitate the design of deep networks for new tasks. Existing techniques rely on two stages: searching over the architecture space and validating the best architecture. NAS algorithms are currently compared solely based on their results on the downstream task. While intuitive, this fails to explicitly evaluate the effectiveness of their search strategies. In this paper, we propose to evaluate the NAS search phase. To this end, we compare the quality of the solutions obtained by NAS search policies with that of random architecture selection. We find that: (i) On average, the state-of-the-art NAS algorithms perform similarly to the random policy; (ii) the widely-used weight sharing strategy degrades the ranking of the NAS candidates to the point of not reflecting their true performance, thus reducing the effectiveness of the search process. We believe that our evaluation framework will be key to designing NAS strategies that consistently discover architectures superior to random ones.

## 1 Introduction

By automating the design of a neural network for the task at hand, Neural Architecture Search (NAS) has tremendous potential to impact the practicality of deep learning (Zoph & Le, 2017; Liu et al., 2018b;a; Tan et al., 2018; Baker et al., 2016), and has already obtained state-of-the-art performance on many tasks. A typical NAS technique (Zoph & Le, 2017; Pham et al., 2018; Liu et al., 2018a) has two stages: the search phase, which aims to find a good architecture, and the evaluation one, where the best architecture is trained from scratch and validated on the test data.

In the literature, NAS algorithms are typically compared based on their results in the evaluation phase. While this may seem intuitive, the search phase of these algorithms often differ in several ways, such as their architecture sampling strategy and the search space they use, and the impact of these individual factors cannot be identified by looking at the downstream task results only. Furthermore, the downstream task results are often reported for a single random seed, which leaves unanswered the question of robustness of the search strategies.

Table 1: **Comparison of NAS algorithms with random sampling.** We report results on PTB using mean validation perplexity (the lower, the better) and on CIFAR-10 using mean top 1 accuracy. We also provide the *p-value* of Student's t-tests against random sampling.

|        | PTB (PPL)        | t-test | CIFAR-10 (acc.)  | t-test |
|--------|------------------|--------|------------------|--------|
| ENAS   | $59.88 \pm 1.92$ | 0.73   | $96.79 \pm 0.11$ | 0.01   |
| DARTS  | $60.61 \pm 2.54$ | 0.62   | $96.62 \pm 0.23$ | 0.20   |
| NAO    | $61.99 \pm 1.95$ | 0.02   | $96.86 \pm 0.17$ | 0.00   |
| Random | $60.13 \pm 0.65$ | -      | $96.44 \pm 0.19$ | -      |

---

[*]Equal contribution
[†]Work done at Swisscom digital lab.

In this paper, we therefore propose to investigate the search phase of existing NAS algorithms in a controlled manner. To this end, we compare the quality of the NAS solutions with a random search policy, which uniformly randomly samples an architecture from the same search space as the NAS algorithms, and then trains it using the same hyper-parameters as the NAS solutions. To reduce randomness, the search using each policy, i.e., random and NAS ones, is repeated several times, with different random seeds.

We perform a series of experiments on the Penn Tree Bank (PTB) (Marcus et al., 1994a) and CIFAR-10 (Krizhevsky et al., 2009) datasets, in which we compared the state-of-the-art NAS algorithms whose code is publicly available—DARTS (Liu et al., 2019b), NAO (Luo et al., 2018) and ENAS (Pham et al., 2018)—to our random policy. We reached the surprising conclusions that, as shown in Table 1, none of them significantly outperforms random sampling. Since the mean performance for randomly-sampled architectures converges to the mean performance over the entire search space, we further conducted *Welch Student's t-tests (Welch, 1947)*, which reveal that, in RNN space, ENAS and DARTS cannot be differentiated from the mean of entire search space, while NAO yields worse performance than random sampling. While the situation is slightly better in CNN space, all three algorithms still perform similarly to random sampling. Note that this does not necessarily mean that these algorithms perform poorly, but rather that the search space has been sufficiently constrained so that even a random architecture in this space provides good results. To verify this, we experiment with search spaces where we can exhaustively evaluate all architectures, and observe that these algorithms truly cannot discover top-performing architectures.

In addition to this, we observed that the ranking by quality of candidate architectures produced by the NAS algorithms during the search does not reflect the true performance of these architectures in the evaluation phase. Investigating this further allowed us to identify that weight sharing (Pham et al., 2018), widely adopted to reduce the amount of required resources from thousands of GPU days to a single one, harms the individual networks' performance. More precisely, using reduced search spaces, we make use of the Kendall Tau $\tau$ metric[1] to show that the architecture rankings obtained with and without weight sharing are entirely uncorrelated in RNN space ($\tau$ = -0.004 over 10 runs); and have little correlation in the CNN space ($\tau$ = 0.195 over 10 runs). Since such a ranking is usually treated as training data for the NAS sampler in the search phase, this further explains the small margin between random search and the NAS algorithms. We also show that training samplers without weight sharing in CNN space surpasses random sampling by a significant margin.

In other words, we disprove the common belief that the quality of architectures trained with and without weight sharing is similar. We show that the difference in ranking negatively impacts the search phase of NAS algorithms, thus seriously impeding their robustness and performance.

In short, evaluating the search phase of NAS, which is typically ignored, allowed us to identify two key characteristics of state-of-the-art NAS algorithms: The importance of the search space and the negative impact of weight sharing. We believe that our evaluation framework will be instrumental in designing NAS search strategies that are superior to the random one. Our code is publicly available at `https://github.com/kcyu2014/eval-nas`.

## 2 RELATED WORK

Since its introduction in (Zoph & Le, 2017), NAS has demonstrated great potential to surpass the human design of deep networks for both visual recognition (Liu et al., 2018b; Ahmed & Torresani, 2018; Chen et al., 2018; Pérez-Rúa et al., 2018; Liu et al., 2019a) and natural language processing (Zoph & Le, 2017; Pham et al., 2018; Luo et al., 2018; Zoph et al., 2018; Liu et al., 2018b; Cai et al., 2018a). Existing search strategies include reinforcement learning (RL) samplers (Zoph & Le, 2017; Zoph et al., 2018; Pham et al., 2018), evolutionary algorithms (Xie & Yuille, 2017; Real et al., 2017; Miikkulainen et al., 2019; Liu et al., 2018b; Lu et al., 2018), gradient-descent (Liu et al., 2019b), bayesian optimization (Kandasamy et al., 2018; Jin et al., 2019; Zhou et al., 2019) and performance predictors (Liu et al., 2018a; Luo et al., 2018). Here, our goal is not to introduce a new search policy, but rather to provide the means to analyze existing ones. Below, we briefly discuss existing NAS methods and focus on how they are typically evaluated.

**Neural architecture search with weight sharing.**   The potential of vanilla NAS comes with the drawback of requiring thousands of GPU hours even for small datasets, such as PTB and CIFAR-10.

---

[1]The Kendall Tau (Kendall, 1938) metric measures the correlation of two ranking. Details in Appendix A.1.

Furthermore, even when using such heavy computational resources, vanilla NAS has to restrict the number of trained architectures from a total of $10^9$ to $10^4$, and increasing the sampler accuracy can only be achieved by increasing the resources.

ENAS (Pham et al., 2018) was the first to propose a training scheme with shared parameters, reducing the resources from thousands of GPU days to one. Instead of being trained from scratch each sampled model inherits the parameters from previously-trained ones. Since then, NAS research has mainly focused on two directions: 1) Replacing the RL sampler with a better search algorithm, such as gradient descent (Liu et al., 2019b), bayesian optimiziation (Zhou et al., 2019) and performance predictors (Luo et al., 2018); 2) Exploiting NAS for other applications, e.g., object detection (Ghiasi et al., 2019; Chen et al., 2019), semantic segmentation (Liu et al., 2019a), and finding compact networks (Cai et al., 2018b; Wu et al., 2018; Chu et al., 2019; Guo et al., 2019).

**Characterizing the search space.** Ying et al. (2019); Dong & Yang (2020) introduced a dataset that contains the ground-truth performance of CNN cells, and Wang et al. (2019) evaluated some traditional search algorithms on it. Similarly, Radosavovic et al. (2019) characterizes many CNN search spaces by computing the statistics of a set of sampled architectures, revealing that, for datasets such as CIFAR-10 or ImageNet, these statistics are similar. While these works support our claim that evaluation of NAS algorithms is crucial, they do not directly evaluate the state-of-the-arts NAS algorithms as we do here.

**Evaluation of NAS algorithms.** Typically, the quality of NAS algorithms is judged based on the results of the final architecture they produce on the downstream task. In other words, the search and robustness of these algorithms are generally not studied, with (Liu et al., 2019b; So et al., 2019) the only exception for robustness, where results obtained with different random seeds were reported. Here, we aim to further the understanding of the mechanisms behind the search phase of NAS algorithms. Specifically, we propose doing so by comparing them with a simple random search policy, which uniformly randomly samples one architecture per run in the same search space as the NAS techniques.

While some works have provided partial comparisons to random search, these comparisons unfortunately did not give a fair chance to the random policy. Specifically, (Pham et al., 2018) reports the results of only a single random architecture, and (Liu et al., 2018b) those of an architecture selected among 8 randomly sampled ones as the most promising one after training for 300 epochs only. Here, we show that a fair comparison to the random policy, obtained by training all architectures, i.e., random and NAS ones, for 1000 epochs and averaging over multiple random seeds for robustness, yields a different picture; the state-of-the-art search policies are no better than the random one.

The motivation behind this comparison was our observation of only a weak correlation between the performance of the searched architectures and the ones trained from scratch during the evaluation phase. This phenomenon was already noticed by Zela et al. (2018), and concurrently to our work by Li & Talwalkar (2019); Xie et al. (2019); Ying et al. (2019), but the analysis of its impact or its causes went no further. Here, by contrast, we link this difference in performance between the search and evaluation phases to the use of weight sharing.

While this may seem to contradict the findings of Bender et al. (2018), which, on CIFAR-10, observed a strong correlation between architectures trained with and without weight sharing when searching a CNN cell, our work differs from (Bender et al., 2018) in two fundamental ways: 1) The training scheme in (Bender et al., 2018), in which the *entire model* with shared parameters is trained via random path dropping, is fundamentally different from those used by state-of-the-arts weight sharing NAS strategies (Pham et al., 2018; Liu et al., 2019b; Luo et al., 2018); 2) While the correlation in (Bender et al., 2018) was approximated using a small subset of sampled architectures, we make use of a reduced search space where we can perform a complete evaluation of *all* architectures, thus providing an exact correlation measure in this space.

## 3 EVALUATING THE NAS SEARCH

In this section, we detail our evaluation framework for the NAS search phase. As depicted in Fig. 1(a,b), typical NAS algorithms consist of two phases:

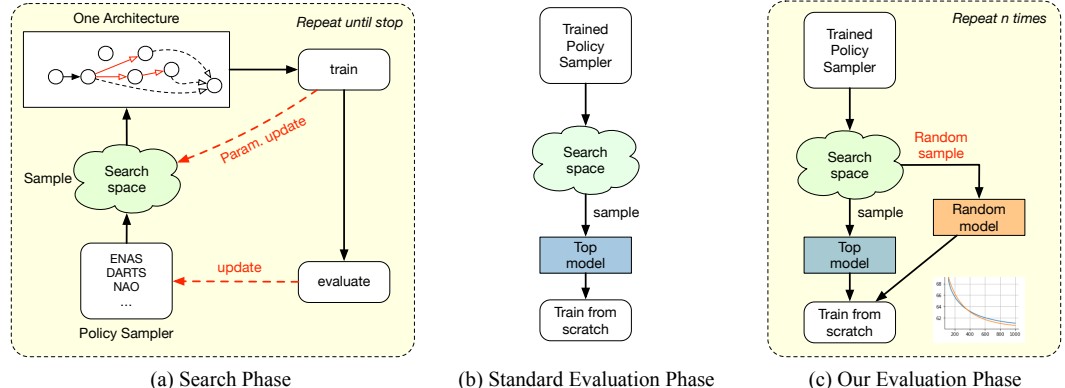

(a) Search Phase      (b) Standard Evaluation Phase      (c) Our Evaluation Phase

Figure 1: **Evaluating NAS.** Existing frameworks consist of two phases: **(a)** The search phase, where a sampler is trained to convergence or a pre-defined stopping criterion; **(b)** The evaluation phase that trains the best model from scratch and evaluates it on the test data. Here, we argue that one should evaluate the search itself. To this end, as shown in **(c)**, we compare the best architecture found by the NAS policy with *a single* uniformly randomly sampled architecture. For this comparison to be meaningful, we repeat it with different random seeds for both training the NAS sampler and our random search policy. We then report the mean and standard deviations over the different seeds.

- **Search:** The goal of this phase is to find the best candidate architecture from the search space[2]. This is where existing algorithms, such as ENAS, DARTS and NAO, differ. Nevertheless, for all the algorithms, the search depends heavily on initialization. In all the studied policies, initialization is random and the outcome thus depends on the chosen random seed.

- **Evaluation:** In this phase, all the studied algorithms retrain the best model found in the search phase. The retrained model is then evaluated on the test data.

The standard evaluation of NAS techniques focuses solely on the final results on the test data. Here, by contrast, we aim to evaluate the search phase itself, which truly differentiates existing algorithms.

To do this, as illustrated in Fig. 1(c), we establish a baseline; we compare the search phase of existing algorithms with a random search policy. An effective search algorithm should yield a solution that clearly outperforms the random policy. Below, we introduce our framework to compare NAS search algorithms with random search. The three NAS algorithms that we evaluated, DARTS (Liu et al., 2019b), NAO (Luo et al., 2018) and ENAS (Pham et al., 2018), are representative of the state of the art for different search algorithms: reinforcement learning, gradient-descent and performance prediction, and are discussed in Appendix C.

### 3.1 COMPARING TO RANDOM SEARCH

We implement our random search policy by simply assigning uniform probabilities to all operations. Then, for each node in the Directed Acyclic Graph (DAG) that is typically used to represent an architecture, we randomly sample a connection to *one* previous node from the resulting distributions.

An effective search policy should outperform the random one. To evaluate this, we compute the validation results of the best architecture found by the NAS algorithm trained from scratch, as well as those of *a single* randomly sampled architecture. Comparing these values for a single random seed would of course not provide a reliable measure. Therefore, we repeat this process for multiple random seeds used both during the search phase of the NAS algorithm and to sample one random architecture as described above. We then report the means and standard deviations of these results over the different seeds. Note that while we use different seeds for the search and random sampling, we always use the same seed when training the models from scratch during the evaluation phase.

Our use of multiple random seeds and of the same number of epochs for the NAS algorithms and for our random search policy makes the comparison fair. This contrasts with the comparisons performed in (Pham et al., 2018), where the results of only *a single random* architecture were reported, and

---

[2]Details about search spaces are provided in Appendix B.

in (Liu et al., 2019b), which selected a single best random architecture among an initial set of 8 after training for 300 epochs only. As shown in Appendix D.2, some models that perform well in the early training stages may yield worse performance than others after convergence. Therefore, choosing the best random architecture after only 300 epochs for PTB and 100 for CIFAR-10, and doing so for a single random seed, might not be representative of the general behavior.

## 3.2 SEARCH IN A REDUCED SPACE

Because of the size of standard search spaces, one cannot understand the quality of the search by fully evaluating all possible solutions. Hence, we propose to make use of reduced search spaces with ground-truth architecture performances available to evaluate the search quality. For RNNs, we simply reduce the number of nodes in the search space from 12 to 2. Given that each node is identified by two values, the ID of the incoming node and the activation function, the space has a cardinality $|S| = n! * |\mathcal{O}|^n$, where $n = 2$ nodes and $|\mathcal{O}| = 4$ operations, thus yielding 32 possible solutions. To obtain ground truth, we train all of these architectures individually. Each architecture is trained 10 times with a different seed, which therefore yields a mean and standard deviation of its performance. The mean value is used as ground truth—the actual potential of the given architecture. These experiments took around 5000 GPU hours.

For CNNs, we make use of NASBench-101 (Ying et al., 2019), a CNN graph-based search space with 3 possible operations, conv3x3, conv1x1 and max3x3. This framework defines search spaces with between 3 and 7 nodes, with 423,624 architectures in 7-node case. To the best of our knowledge, we are the first to evaluate the NAS methods used in this paper on NASBench.

## 4 EXPERIMENTAL RESULTS

To analyze the search phase of the three state-of-the-art NAS algorithms mentioned above, we first compare these algorithms to our random policy when using standard search spaces for RNNs on (PTB) and CNNs on CIFAR-10. Details about the experiment setting are in Appendix C.5. The surprising findings in this typical NAS use case prompted us to study the behavior of the search strategies in reduced search spaces. This allowed us to identify a factor that has a significant impact on the observed results: Weight sharing. We then quantify this impact on the ranking of the NAS candidates, evidencing that it dramatically affects the effectiveness of the search.

## 4.1 NAS COMPARISON IN A STANDARD SEARCH SPACE

Below, we compare DARTS (Liu et al., 2019b), NAO (Luo et al., 2018), ENAS (Pham et al., 2018) and BayesNAS (Zhou et al., 2019) with our random search policy, as discussed in Section 3.1. We follow (Liu et al., 2019b) to define an RNN search space of 12 nodes and a CNN ones of 7 nodes. For each of the four search policies, we run 10 experiments with a different initialization of the sampling policy. During the search phase, we used the authors-provided hyper-parameters and code for each policy. Once a best architecture is identified by the search phase, it is used for evaluation, i.e., we train the chosen architecture from scratch for 1000 epochs for RNN and 600 for CNN.

**RNN Results.** In Figure 2, we plot, on the left, the mean perplexity evolution over the 1000 epochs, obtained by averaging the results of the best architectures found using the 10 consecutive seeds.[3] On the right, we show the perplexity evolution for the best cell of each strategy among the 10 different runs. Random sampling is robust and consistently competitive. As shown in Table 1, it outperforms on average the DARTS and NAO policies, and yields the overall best cell for these experiments with perplexity

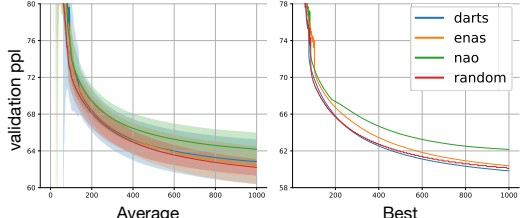

Figure 2: Validation perplexity evolution in the 12-node RNN search space. (Best viewed in color)

---

[3]Starting from 1268, which is right after 1267, the seed released by Liu et al. (2019b). Note that, using this seed, we can reproduce the DARTS RNN search and obtain a validation PPL of 55.7 as in (Liu et al., 2019b).

of 57.60. Further training this cell for 4000 epochs, as in (Liu et al., 2019b), yields a perplexity of 55.93. The excellent performance of the random policy evidences the high expressiveness of the manually-constructed search space; even arbitrary policies in this space perform well, as evidenced by the relatively low standard deviation over the 10 seeds of the random architectures, shown in Table 1 and Figure 2(left).

**CNN Results.** In Table 2, we compare the NAS methods with our random policy in the search space of Liu et al. (2019b). We provide the accuracy reported in the original papers as well as the accuracy we reproduced using our implementation. Note that the NAS algorithms only marginally outperform random search, by less than 0.5% in top-1 accuracy. The best architecture was discovered by NAO, with an accuracy of 97.10%, again less than 0.5% higher than the randomly discovered one. Note that, our random sampling comes at no search cost. By contrast, Li & Talwalkar (2019) obtained an accuracy of 97.15% with a different random search policy having the same cost as DARTS.

Table 2: **Top 1 accuracy in the 7-node DARTS Search space.** We report the mean and best top-1 accuracy on the test sets of architectures found by DARTS, NAO, ENAS, BayesNAS, and our random policy. As sanity check, we also train from scratch the architectures reported in original papers, as well as their reported performance.

| | Our seed | | Best reported result | |
|---|---|---|---|---|
| Type | Mean test | Best test | Original | Reproduced |
| DARTS | $96.62 \pm 0.23$ | 96.80 | **97.24** | **97.15** |
| NAO | **$96.86 \pm 0.17$** | **97.10** | 96.47 | 96.92 |
| ENAS | $96.76 \pm 0.10$ | 96.95 | 96.46 | 96.87 |
| BayesNAS | $95.99 \pm 0.25$ | 96.41 | 97.19 | 97.13 |
| Random | $96.48 \pm 0.18$ | 96.74 | 97.15 [†] | |

[†]Result took from Li & Talwalkar (2019)

**Observations:**

- The evaluated state-of-the-art NAS algorithms do not surpass random search by a significant margin, and even perform worse in the RNN search space.
- The ENAS policy sampler has the lowest variance among the three tested ones. This shows that ENAS is more robust to the variance caused by the random seed of the search phase.
- The NAO policy is more sensitive to the search space; while it yields the best performance in CNN space, it performs the worst in RNN one.
- The DARTS policy is very sensitive to random initialization, and yields the largest standard deviation across the 10 runs (2.54 in RNN and 0.23 in CNN space).

Such a comparison of search policies would not have been possible without our framework. Nevertheless, the above analysis does not suffice to identify the reason behind these surprising observations. As mentioned before, one reason could be that the search space has been sufficiently constrained so that all architectures perform similarly well. By contrast, if we assume that the search space does contain significantly better architectures, then we can conclude that these search algorithms truly fail to find a good one. To answer this question, we evaluate these methods in a reduced search space, where we can obtain the true performance of all possible architectures.

## 4.2 Searching a Reduced Space

The results in the previous section highlight the inability of the studied methods to surpass random search. Encouraged by these surprising results, we then dig deeper into their causes. Below, we make use of search spaces with fewer nodes, which we can explore exhaustively.

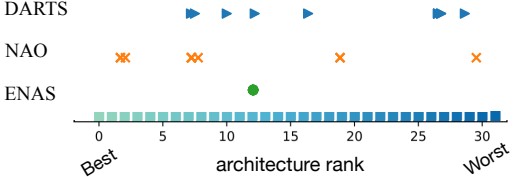

**Reduced RNN space.** We use the same search space as in Section 3.2 but reduce the number of intermediate nodes to 2. In Table 3 (A), we provide the results of searching the RNN 2-node space. Its smaller size allows us to exhaustively compute the results of all possible solutions, thus determining the upper bound for this case. In Figure 3, we plot the rank of the top 1 architecture discovered by the three NAS algorithms for each of the 10 different runs.

Figure 3: **Architectures discovered by NAS algorithms.** We rank all 32 architectures in the reduced search space based on their performance of individual training, from left (best) to right (worst), and plot the best cell found by three NAS algorithms across the 10 random seeds.

Table 3: **Results in reduced search spaces.** For RNNs (A), We report the mean and best perplexity on the validation and test sets at the end of training the architectures found using DARTS, NAO, ENAS. For CNNs (B), we show the mean and best top-1 accuracy on the test set. Instead of running random sampling in the reduced space, we compute the probability of the best model found by each method to surpass the random one (details in Appendix A.2). The mean and best statistics of the entire search space are reported as Space.

| Type | (A) **RNN** n=2(32) in PPL. | | | | (B) **NASBench** n=7(423K) | | | |
|------|------------|-----------|------------|-----------|-----------|-----------|-----------|------------|
| | Mean Valid | Mean Test | Best Valid | Best Test | Mean Acc. | Best Acc. | Best Rank | p(>random) |
| DARTS | $71.29 \pm 2.45$ | $68.74 \pm 2.42$ | 68.05 | 65.55 | $92.21 \pm 0.61$ | 93.02 | 57079 | 0.24 |
| NAO | $\mathbf{68.66 \pm 2.50}$ | $\mathbf{66.03 \pm 2.40}$ | **66.22** | **63.59** | $\mathbf{92.59 \pm 0.59}$ | **93.33** | **19552** | **0.62** |
| ENAS | $69.99 \pm 0.0$ | $66.61 \pm 0.0$ | 69.99 | 66.61 | $91.83 \pm 0.42$ | 92.54 | 96939 | 0.07 |
| Space | $69.69 \pm 2.44$ | $67.21 \pm 2.52$ | 65.38 | 62.63 | $90.93 \pm 5.84$ | 95.06 | - | - |

We observe that: (i) All policies failed to find the architecture that actually performs best; (ii) The ENAS policy always converged to the same architecture. This further evidences the robustness of ENAS to the random seed; (iii) NAO performs better than random sampling on average because it keeps a ranking of architectures; (iv) DARTS never discovered a top-5 architecture.

**Reduced CNN space.** In Table 3 (B), we report the mean and best test top-1 accuracy over 10 different runs on the NASBench-101 7-node space. To assess the search performance, we also show the best architecture rank in the entire space. The best test accuracy found by these methods is 93.33, by NAO, which remains much lower than the ground-truth best of 95.06. In terms of ranking, the best rank of these methods across 10 runs is 19522, which is among the top 4% architectures and yields a probability of 0.62 to surpass a randomly-sampled one given the same search budget. Note that ENAS and DARTS only have 7% and 24% chance to surpass the random policy. See Appendix A.2 for the definition of this probability, and Appendix D.3 for detailed results.

NAO seems to constantly outperform random search in the reduced space. Nevertheless, the final architecture chosen by NAO is *always* one of the architectures from the initial pool, which were sampled uniformly randomly. This indicates that the ranking of NAO is not correctly updated throughout the search and that, in practice, in a reduced space, NAO is similar to random search.

## 4.3 IMPACT OF WEIGHT SHARING

Our previous experiments in reduced search spaces highlight that the ranking of the searched architectures does not reflect the ground-truth one. As we will show below, this can be traced back to weight sharing, which all the tested algorithms, and the vast majority of existing ones, rely on. To evidence this, we perform the following experiments:

**Without WS:** We make use of the reduced space, where we have the architecture's real performance.

**With WS:** We train the architectures in parallel, using the weight sharing strategy employed in NAO and ENAS. As DARTS does not have discrete representations of the solutions during the search, the idea of solution ranking does not apply. During training, each mini-batch is given to an architecture uniformly sampled from the search space. We repeat the process 10 times, with 10 random seeds and train the shared weights for 1000 epochs for the RNN experiments and 200 epochs for the CNN ones. Note that, this approach is equivalent to Single Path One Shot (SPOS) (Guo et al., 2019). It guarantees equal expectations of the number of times each architecture is sampled, thus overcoming the bias due to unbalanced training resulting from ineffective sampling policies.

We then compute the correlation between the architecture rankings found with WS and the ground truth (i.e., the architectures trained independently). For each of the 10 runs of the weight sharing strategy, we evaluate the Kendall Tau metric (defined in Appendix A.1) of the final rankings with respect to the real averaged ranking.

**RNN Results.** In Figure 4(a), we depict the architecture performance obtained without WS (sorted in ascending order of average validation perplexity), and the corresponding performance with WS. In Figure 4(b), we show the rank difference, where the best and worst were found using the Kendall Tau metric, and show a concrete rank change example in Figure 4(c).

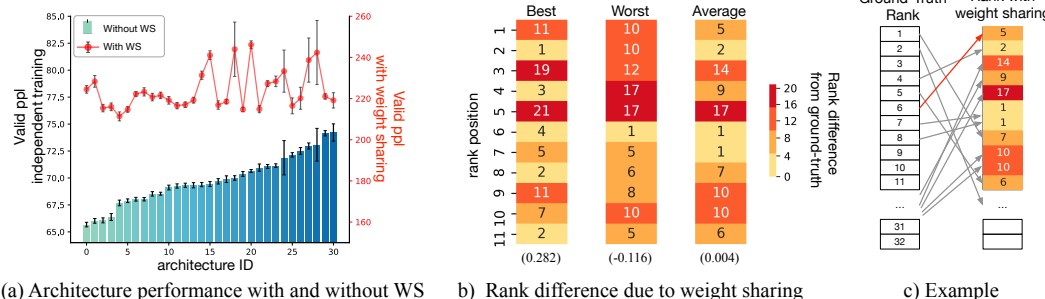

(a) Architecture performance with and without WS    b) Rank difference due to weight sharing    c) Example

Figure 4: **Rank disorder due to weight sharing in RNN reduced space. (a)** We report the average and std over 10 different runs. Note that the rankings significantly differ between using (line plot) and not using WS (bar plot), showing the negative impact of this strategy. **(b)** We visualize from left to right, the best, worst and average cases, and show the corresponding Kendall Tau value. A change in ranking, indicated by the colors and numbers, is measured as the absolute position change between the WS ranking and the true one. For conciseness, we only show the top 10 architectures. **(c)** For example, in the average scenario, the 6-th best architecture is wrongly placed as the best one, as indicated by the red arrow.

**CNN Results.** We report the average Kendall tau across 10 different runs. Note that we sampled up to 200 architectures for each experiment and fully evaluated on the entire test set to use the test accuracy for ranking. The Kendall tau for search spaces from 3 to 7 nodes is, respectively, 0.441, 0.314, 0.214, 0.195. We also provide other statistics in Table 6 of Appendix D.3.

Table 4: **Search results w/o weight sharing.** We report results from ENAS ans NAO on NASBench with 7 nodes over 10 runs.

| Type | Mean Acc. | Best Acc. | Best Rank | P(>random) |
|------|-----------|-----------|-----------|------------|
| NAO | $93.08 \pm 0.71$ | 94.11 | 3543 | 0.92 |
| ENAS | $93.54 \pm 0.45$ | 94.04 | 4610 | 0.90 |

Since NAO and ENAS intrinsically disentangle the training of shared weights and sampler, to further confirm the negative effect of weight sharing, we adapt these algorithms to use the architecture's performance in the NASBench dataset to train their sampler. Table 4 evidences that, after removing weight sharing, both ENAS and NAO consistently discover a good architecture, as indicated by a small difference between the best over 10 runs and the mean performance. More interestingly, for the 7-node case, the best cell discovered (94.11% by NAO and 94.04% by ENAS) are more than 1% higher than the best cells found with weight sharing (93.33 and 92.54, respectively, in Table 3).

**Observations:**

- The difference of architecture performance is *not* related to the use of different random seeds, as indicated by the error bars in Figure 4(a).

- WS *never* produces the true ranking, as evidenced by the Best case in Figure 4(b).

- The behavior of the WS rankings is greatly affected by changing the seed. In particular, the Kendall Tau for the plots in Figure 4(b) are 0.282, −0.004, −0.116 for Best, Average and Worst.

- For RNNs, the Kendall Tau are close to 0, which suggests a lack of correlation between the WS rankings and the true one. By contrast, for CNNs, the correlation is on average higher than for RNNs. This matches the observation in Section 4.1 that CNN results are generally better than RNN ones.

- In a reduced CNN space, the ranking disorder increases with the space complexity, i.e., this disorder is proportional to the amount of weight sharing.[4]

- If we train NAO and ENAS without weight sharing in NASBench, on average the performance is 1% higher than them with it. This further evidences that weight sharing negatively impacts the sampler, and with a good ranking, the sampler can be trained better. Furthermore, the probability to surpass random search increases from 0.62 to 0.92 for NAO and from 0.07 to 0.90 for ENAS.

---

[4]We also conduct another experiment regarding to the amount of sharing in Appendix D.1

Together with previous results, we believe that these results evidence the negative impact of weight sharing; it dramatically affects the performance of the sampled architectures, thus complicating the overall search process and leading to search policies that are no better than the random one.

## 5 CONCLUSION

In this paper, we have analyzed the effectiveness of the search phase of NAS algorithms via fair comparisons to random search. We have observed that, surprisingly, the search policies of state-of-the-art NAS techniques are no better than random, and have traced the reason for this to the use of (i) a constrained search space and (ii) weight sharing, which shuffles the architecture ranking during the search, thus negatively impacting it.

In essence, our gained insights highlight two key properties of state-of-the-art NAS strategies, which had been overlooked in the past due to the single-minded focus of NAS evaluation on the results on the target tasks. We believe that this will be key to the development of novel NAS algorithms. In the future, we will aim to do so by designing relaxed weight sharing strategies.

## 6 ACKNOWLEDGEMENT

This work was supported in part by the Swiss National Science Foundation. We would also like to thank Rene Ranftl and Vladlen Koltun for the discussions and support.

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

## A    METRICS TO EVALUATE NAS ALGORITHMS

### A.1    KENDALL TAU METRIC

As a correlation measure, we make use of the *Kendall Tau ($\tau$)* metric (Kendall, 1938): a number in the range [-1, 1] with the following properties:

- $\tau = -1$: Maximum disagreement. One ranking is the opposite of the other.
- $\tau = 1$: Maximum agreement. The two rankings are identical.
- $\tau$ close to 0: A value close to zero indicates the absence of correlation.

### A.2    PROBABILITY TO SURPASS RANDOM SEARCH

As discussed in Section 3.2, the goal of NASBench is to search for a CNN cell with up to 7 nodes and 3 operations, resulting in total 423,624 architectures. Each architecture is trained 3 times with different random initialization up to 108 epochs on the CIFAR-10 training set, and evaluated on the test split. Hence, the average test accuracy of these runs can be seen as the ground-truth performances. In our experiments, we use this to rank the architectures, from 1 (highest accuracy) to 423,624. Given the best architecture's rank $r$ after $n$ runs, and maximum rank $r_{max}$ equals to the total number of architectures, the probability that the best architecture discovered is better than a randomly searched one given the same budget is given by

$$p = 1 - (1 - (r/r_{max}))^n. \tag{1}$$

We use this as a new metric to evaluate the search phase.

## B    NAS SEARCH SPACE REPRESENTATION

As discussed in the main paper, our starting point is a neural search space for a neural architecture, as illustrated in Figure 5. A convolutional cell can be represented with a similar topological structures. Following common practice in NAS (Zoph & Le, 2017), a candidate architecture sampled from this space connects the input and the output nodes through a sequence of intermediary ones. Each node is connected to others and has an operation attached to it.

A way of representing this search space (Pham et al., 2018; Luo et al., 2018), depicted in Figure 5(b), is by using strings. Each character in the string indicates either the node ID that the current node is connected to, or the operation selected for the current node. Operations include the identity, sigmoid, tanh and ReLU (Nair & Hinton, 2010).

Following the alternative way introduced in (Liu et al., 2019b), we make use of a vectorized representation of these strings. More specifically, as illustrated by Figure 5(c), a node ID, resp. an operation, is encoded as a vector of probabilities over all node IDs, resp. all operations. For instance, the connection between nodes $i$ and $j$ is represented as $y^{(i,j)}(x) = \sum_{o \in \mathcal{O}} p_o o(x)$, with $\mathcal{O}$ the set of all operations, and $p_o = \texttt{softmax}(\alpha_o) = \exp(\alpha_o)/\sum_{o' \in \mathcal{O}} \exp(\alpha_{o'})$ the probability of each operation.

## C    NAS ALGORITHMS

Here, we discuss the three state-of-the-art NAS algorithms used in our experiments in detail, including their hyper-parameters during the search phase. The current state-of-the-arts NAS on CIFAR-10 is ProxylessNAS (Cai et al., 2018b) with a top-1 accuracy of 97.92. However, this algorithm inherits the sampler from ENAS and DARTS, but with a different objective function, backbone model, and search space. In addition, the code is not publicly available, which precludes us from directly evaluating it.

### C.1    ENAS

adopts a reinforcement learning sampling strategy that is updated with the REINFORCE algorithm. The sampler is implemented as a two-layer LSTM Hochreiter & Schmidhuber (1997) and generates a sequence of strings. In the training process, each candidate sampled by the ENAS controller is trained on an individual mini-batch. At the end of each epoch, the controller samples new architectures

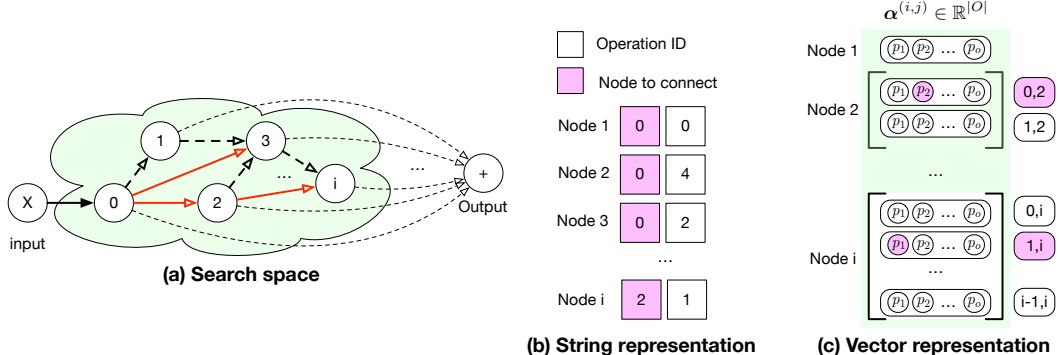

Figure 5: **Search space of NAS algorithms.** Typically, the search space is encoded as **(a)** a directed acyclic graph, and an architecture can be represented as **(b)** a string listing the node ID that each node is connected to, or the operation ID employed by each node. **(c)** An alternatively representation is a list of vectors $\alpha$ of size $\frac{n(n+1)}{2}|\mathcal{O}|$, where $n$ is the number of nodes and $\mathcal{O}$ is the set of all operations. Each vector, $\alpha^{(i,j)}$, captures, via a softmax, the probability $p_o$ that operation $o$ is employed between node $i$ and $j$. Note that any node only takes one incoming edge, thus (b) and (c) represent the same search space and only differs in its formality.

that are evaluated on a single batch of the validation dataset. After this, the controller is updated accordingly using these validation metrics. We refer the reader to (Pham et al., 2018) for details about the hyper-parameter settings.

## C.2  DARTS

It vectorizes the aforementioned strings as discussed in Section B and shown in Fig. 5(c). The sampling process is then parameterized by the vector $\alpha$, which is optimized via gradient-descent in a dual optimization scheme: The architecture is first trained while fixing $\alpha$, and $\alpha$ is then updated while the network is fixed. This process is repeated in an alternating manner. In the evaluation phase, DARTS samples the top-performing architecture by using the trained $\alpha$ vector as probability prior, i.e., the final model is not a soft average of all paths but one path in the DAG, which makes its evaluation identical to that of the other NAS algorithms. Note that we use the same hyper-parameters as in the released code of Liu et al. (2019b).

## C.3  NAO

It implements a gradient-descent algorithm, but instead of vectorizing the strings as in DARTS, it makes use of a variational auto-encoder (VAE) to learn a latent representation of the candidate architectures. Furthermore, it uses a performance predictor, which takes a latent vector as input to predict the corresponding architecture performance. In short, the search phase of NAO consists of first randomly sampling an initial pool of architectures and training them so as to obtain a ranking. This ranking is then used to train the encoder-predictor-decoder network, from which new candidates are sampled, and the process is repeated in an iterative manner. The best architecture is then taken as the top-1 in the NAO ranking. We directly use the code released by Luo et al. (2018).

## C.4  BAYESNAS

Bayesian optimization was first introduced to the neural architecture search field by Kandasamy et al. (2018) and Jin et al. (2019). We chose to evaluate BayesNAS (Zhou et al., 2019) because it is more recent than Auto-Keras (Jin et al., 2019) and than the work of Kandasamy et al. (2018), and because these two works use different search spaces than DARTS, resulting in models with significantly worse performance than DARTS. BayesNAS adopts Bayesian optimization to prune the fully-connected DAG graph using the shared weights to obtain accuracy metrics. The search space follows that of DARTS (Liu et al., 2019b) with minor modifications in connections, but exactly the same operations. Please see (Zhou et al., 2019) for more details. Note that BayesNAS was only implemented in CNN

space. We use the search and model code released by Zhou et al. (2019) with our training pipeline, since the authors did not release the training code.

### C.5 EXPERIMENTAL SETUP

Following common practice in NAS, we make use of the word-level language modeling Penn Tree Bank (PTB) dataset (Marcus et al., 1994b) and of the image classification CIFAR-10 dataset (Krizhevsky et al., 2009). For these datasets, the goals are, respectively, finding a recurrent cell that correctly predicts the next word given the input sequence, and finding a convolutional cell that maximizes the classification accuracy. The quality of a candidate is then evaluated using the *perplexity* metric and top-1 accuracy, respectively.

In the evaluation phase, we always use the same model backbone and parameter initialization for all searched architectures, which ensures fairness and reflects the empirical observation that the searched models are insensitive (accuracy variations of less than 0.002 (Liu et al., 2019b)) to initialization during evaluation. For our RNN comparisons, we follow the procedure used in (Liu et al., 2019b; Pham et al., 2018; Luo et al., 2018) for the final evaluation, consisting of keeping the connections found for the best architecture in the search phase but increasing the hidden state size (to 850 in practice), so as to increase capacity. Furthermore, when training an RNN architecture from scratch, we follow (Yang et al., 2017; Merity et al., 2017) and first make the use of standard SGD to speed up training, and then change to average SGD to improve convergence. For all CNN architectures, we use RMSProp for fast optimization (Ying et al., 2019) and enable auxiliary head and cut-out (DeVries & Taylor, 2017) to boost the performance as in Liu et al. (2019b).

### C.6 ADAPTATION TO REDUCED SEARCH SPACE

When changing to reduced search spaces, we adapted the evaluated search algorithms to achieve the best performance. Below, we describe these modifications.

**RNN reduced space**

- For DARTS, no changes are needed except modifying the number of nodes in the search space.
- For NAO, to mimic the behavior of the algorithm in the space of 12 nodes, we randomly sample 20% of the possible architectures to define the initial candidate pool. We train the encoder-predictor-decoder network for 250 iterations every 50 epochs using the top-4 architectures in the NAO ranking. At each search iteration, we sample at most 3 new architectures to be added to the pool. The rest of the search logic remains unchanged.
- For ENAS, we reduce the number of architectures sampled in one epoch to 20 and increase the number of batches to 10 for each architecture. All other hyper-parameters are unchanged.

**CNN reduced space**

- For DARTS, again, no changes are needed except modifying the number of nodes in the search space.
- For NAO, since the topology of the NASBench space is very similar to the original search space, we kept most of the parameters unchanged, but only change the embedding size of the encoder proportionally to the number of nodes ($12 \times$ node - 12).
- For ENAS, we set the LSTM sampler size to 64 and keep the temperature as 5.0. The number of aggregation step of each sampler training is set to 10.

## D    SUPPLEMENTARY EXPERIMENTS

We provide additional experiments to support our claims.

### D.1 INFLUENCE OF THE AMOUNT OF SHARING

Depending on the active connections in the DAG, different architectures are subject to different amounts of weight sharing. In Figure 6 (a), let us consider the 3-node case, with node 1 and node 2

fixed and node 3 having node 1 as incoming node. In this scenario, the input to node 3 can be either directly node 0 (i.e., the input), or node 1, or node 2. In the first case, the only network parameters that the output of node 3 depends on are the weights of its own operation. In the second and third cases, however, the output further depends on the parameters of node 1, and of nodes 1 and 2, respectively.

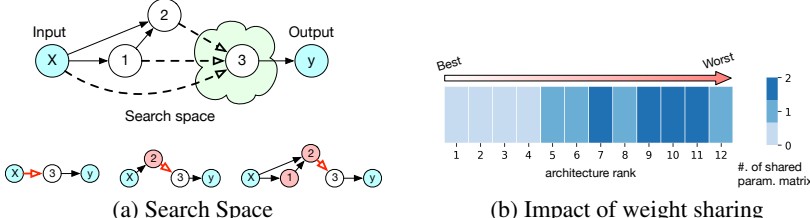

(a) Search Space         (b) Impact of weight sharing

Figure 6: **A toy search space to assess the influence of weight sharing in RNN space.**
(a) Top: the reduced space. (a) Bottom: The amount of sharing depends on the activated path.
(b) Ranking obtained from weight sharing training, that best ranked architectures has weight matrix to share.

To study the influence of the amount of sharing on the architecture ranking, we performed an experiment where we fixed the first two nodes and only searched for the third one. This represents a space of 12 architectures (3 possible connections to node 3 × 4 operations). We train them using the same setting in Section 4.3. The ranking of the 12 architectures is shown in Figure 6 (b), where color indicates the number of shared weight matrices, that is, matrices of nodes 1 and 2 also used in the search for node 3. Note that the top-performing architectures do not share any weights and that the more weights are shared, the worse the architecture performs.

In CNN space, we conduct a similar experiment in NAS-Bench. With total node equals to 6, we only permute the last node operation and connection to one of the previous nodes. In short, we will have a total 4 connection possibility and 3 operation choices, in total 12 architectures. We compute the Kendall Tau among the architectures with the same connection but different operations, and the results are reported in Table 5. Clearly, the correlation of architectures decrease while the weight sharing matrices increase.

Table 5: Ranking disorder of weight sharing in CNN.

| # of shared matrix | 0 | 1 | 2 | 3 |
|---|---|---|---|---|
| Kendall Tau $\tau$ | 0.67. | 0.33 | -0.33 | 0.0 |

### D.2 RANDOM SAMPLING COMPARISON

As discussed before, the random policy in (Liu et al., 2019b) samples 8 architectures, and picks the best after training them for 300 epochs independently. It might seem contradictory that DARTS outperforms this random policy, but cannot surpass the much simpler one designed in our paper, which only randomly samples 10 architectures (1 per random seed), trains them to convergence and picks the best. However, the random policy in DARTS relies on the assumption that a model that performs well in the early training stage will remain effective until the end of training. While this may sound intuitive, we observed a different picture with our reduced search space.

Since we obtained the ground-truth performance ranking, as discussed in Section 4.2 of the main paper, in Figure 7, we plot the evolution of models' rank while training proceeds, based on the average validation perplexity over 10 runs. Clearly, there are significant variations during training: Good models in early stages drop lower in the ranking towards the end. As such, there is a non-negligible chance that the random policy in DARTS

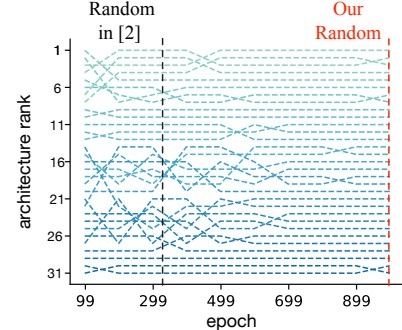

Figure 7: **Rank changes while training.** Each line represents the evolution of the rank of a single architecture. The models are sorted based on their test performance after 1000 epochs, with the best-performing one at the top. The curves were averaged over 10 runs. They correspond to the experiment in Section 4.2. The vertical dashed lines indicate the epoch number where random sampling was performed, either by the random policy in Liu et al. (2019b), or by ours.

Table 6: Comparison of state-of-the-art methods on NASBench-101 search space.

| Search Space | NASBench-101 on CIFAR-10. n: number of nodes, (x): total architecture choices, mean and best: top 1 accuracy (in %) | | | | | | | | | | | | |
|---|---|---|---|---|---|---|---|---|---|---|---|---|
| | n = 4 (91) | | | n = 5 (2.5K) | | | n = 6 (64K) | | | n = 7 (423K) | | | Best of |
| Method | Mean | Best | K-T | Mean | Best | K-T | Mean | Best | K-T | Mean | Best | K-T | all n |
| *Sampling methods, train sampler during training super-net* | | | | | | | | | | | | | |
| ENAS | $89.41 \pm 3.54$ | 92.95 | - | $89.03 \pm 2.76$ | 91.84 | - | $91.41 \pm 1.42$ | 92.75 | - | $91.83 \pm \mathbf{0.42}$ | 92.54 | - | 93.69 |
| NAO | $\mathbf{92.87} \pm 0.69$ | 93.88 | - | $92.07 \pm 1.14$ | **93.97** | - | $92.83 \pm \mathbf{0.78}$ | 93.62 | - | $\mathbf{92.59} \pm 0.59$ | 93.33 | - | 93.97 |
| DARTS | $91.54 \pm 1.93$ | 93.71 | - | $91.82 \pm 1.10$ | 93.63 | - | $91.12 \pm 1.86$ | 93.92 | - | $92.21 \pm 0.61$ | 93.02 | - | 93.92 |
| FBNET | $91.56 \pm 1.89$ | 93.71 | - | $\mathbf{92.51} \pm 1.51$ | 93.90 | - | $91.76 \pm 1.26$ | 92.98 | - | $92.29 \pm 1.25$ | **93.98** | - | 93.98 |
| *One-shot methods, train sampler after optimizing super-net* | | | | | | | | | | | | | |
| SPOS | $91.14 \pm 3.47$ | **94.24** | 0.441 | $91.53 \pm 1.76$ | 93.72 | 0.314 | $90.56 \pm 1.03$ | 92.29 | 0.214 | $89.85 \pm 3.80$ | 93.84 | 0.195 | 94.24 |
| FAIRNAS | $89.08 \pm 4.35$ | 94.13 | -0.043 | $91.38 \pm 1.44$ | 93.55 | -0.028 | $91.75 \pm 2.20$ | **94.47** | -0.221 | $91.10 \pm 1.84$ | 93.55 | -0.232 | **94.47** |

picks a model whose performance will be sub-optimal. We therefore believe that our policy that simply samples one model and trains it until convergence yields a more fair baseline. Furthermore, the fact that we perform our comparison using 10 random seeds, for both our approach and the NAS algorithms, vs a single one in (Liu et al., 2019b) makes our conclusions more reliable.

### D.3 NASBench detailed results.

We provide additional evaluations on the NASBench dataset to benchmark the performance of the state-of-the-art NAS algorithms. In addition to the three methods in the main paper, we re-implemented some recent algorithms, such as FBNet (Wu et al., 2018), Single Path One Shot (SPOS) (Guo et al., 2019), and FairNAS (Chu et al., 2019). Note that we removed the FBNet device look-up table and model latency from the objective function since the search for a mobile model is not our primary goal. This also makes it comparable with the other baselines.

To ensure fairness, after the search phase is completed, each method trains the top 1 architectures found by its policy from scratch to obtain ground-truth performance; we repeated all the experiments with 10 random seeds. We report the mean and best top 1 accuracy in Table 6 for a number of nodes $n \in [4, 7]$, and the Kendall Tau (K-T) values for one-shot methods following Section 4.2 in the paper.

From the results, we observe that: 1) Sampling-based NAS strategies always have better mean accuracy with lower standard deviation, meaning that they converge to a local minimum more easily but do not exploit the entire search space. 2) By contrast, one-shot methods explore more diverse solutions, thus having larger standard deviations but lower means, but are able to pick a better architecture than sampling-based strategies (94.47 for FairNAS and 94.24 for SPOS, vs best of sampler based FBNet 93.98). 3) ENAS constantly improves as the number of nodes increases. 4) FBNet constantly outperforms DARTS, considering the similarity, using Gumbel Softmax seems a better choice. 5) The variance of these algorithms is large and sensitive to initialization. 6) Even one-shot algorithms cannot find the overall best architecture with accuracy 95.06.

