# OpenReview forum: "Evaluating The Search Phase of Neural Architecture Search"
_ICLR.cc/2020/Conference — Accept (Poster)_

### Official Review · AnonReviewer2 · 2019-10-23
**Official Blind Review #2**

**Rating:** 6

**Review:**

This paper studies the effectiveness of several Neural Architecture Search (NAS) methods comparing it with that of random policy search. The paper concludes that none of these methods for a CNN (trained using CIFAR-10) and RNN model (trained using PTB) are statistically significantly better than the random search. The authors suggest that this is due to the weight sharing used by the NAS algorithms to accelerate the network training.

This paper is written well with a good discussion of the problem. The problem considered is important and authors have raised the effectiveness of NAS methods correctly. Before this paper, Li and Talwalkar, “Random Search and Reproducibility for Neural Architecture Search” have also compared some of the NAS methods with random search and reported similar concerns.
In this sense, the paper is not novel although I agree this paper has added an additional insight that “weight sharing” is the culprit.

I have two concerns about the methodology used in this paper:

(1)	The search space has been greatly, just 32 possible architectures. It is well known that in a small search space, difference between the performance of random search and any other systematic search algorithm is quite small. Only when the space gets larger, the power of systematic search starts to show up. Although, I completely understand the authors’ limitation of not having a ground truth for a large search space (infeasible due to a huge computational requirement), but without this, the claim of this paper is weak.

(2)	Secondly, among the NAS methods considered, I missed the whole class of methods based on Bayesian optimization. There are many such work, but I am listing just two of them here: Jin et al. (2018), “AUTO-KERAS: EFFICIENT NEURAL ARCHITECTURE SEARCH WITH NETWORK MORPHISM” and Kandasamy et al. (2018), “Neural Architecture Search with Bayesian Optimisation and Optimal Transport”. It would be useful to have them in the list of NAS methods considered here.

Post Rebuttal:
I have read the rebuttal. I appreciate the authors 'prompt comparison of their method with Bayesian NAS. However, I still think that using a reduced search space, it is not appropriate to compare NAS methods with random search. Moreover, all the claims are only empirical and more experimental evidence needs to be provided to reject the current NAS methods.


**Experience Assessment:**

I have read many papers in this area.

**Review Assessment: Checking Correctness Of Derivations And Theory:**

N/A

**Review Assessment: Checking Correctness Of Experiments:**

I assessed the sensibility of the experiments.

**Review Assessment: Thoroughness In Paper Reading:**

I read the paper at least twice and used my best judgement in assessing the paper.

---

> ### Author Response · Authors · 2019-11-11
> **Response to AnonReviewer2**
>
> Thank you for your time reading our work and providing constructive comments. We provide our responses and hopefully they could resolve your concerns.
>
> Response to your general concern about novelty:
> The focus of Li and Talwalkar and ours are fundamentally different. While they develop a strategy to select more promising models by evaluating the performance of randomly sampled architectures trained with weight sharing, we reveal why state-of-the-art NAS algorithms degrade to random when one considers multiple seeds and not just the one that gives the best results. This is novel and, as acknowledged by the other reviewers, provides valuable insights into the NAS field. Furthermore, Li and Talwalkar only reproduce DARTS (as shown in their paper and code release), and report the statistics for other methods directly from their respective papers. By contrast, here, we took the time to reproduce all baselines in a fair environment to minimize the systematic errors.
>
> Response to methodology concerns:
> (1) We use an RNN reduced space of 32 architectures only to illustrate the downside of weight sharing, not to show that random performs well in such a reduced space. More importantly, the NASBench space we use for the same purpose with CNNs is much larger, consisting of 431K different architectures, which we believe fully validate our claims. Furthermore, the results in CNN space match our observations with RNNs.
>
> (2) Thanks for the advice. We chose to evaluate BayesNAS [1] because it is more recent than Auto-Keras and the work of Kandasamy et al., 2018, and because these two works use different search spaces than DARTS, resulting in models with significantly worse performance than DARTS. We will add a discussion in the updated version.
>
> As for the experiments, the search phase is now finished and we are training the searched models from scratch. This will be done before the rebuttal deadline and we will update the paper accordingly.
>
> --Reference —
> [1] Zhou et al., BayesNAS: A Bayesian Approach for Neural Architecture Search, ICML’19

---

> ### Author Response · Authors · 2019-11-15
> **Updated results for BayesNAS and comments**
>
> The experiments for BayesNAS have now finished. We follow the exact same settings as in Section 4.1 for CNN DARTS space. We used the search code released by the author and the code is posted at https://drive.google.com/file/d/1d8KuUmmnorzia6FVKzNzBHIStqZSmefp/view?usp=sharing
>
> The evaluation code is the same as for the other baselines. Note that the evaluation code is not provided in the released version. In addition, BayseNAS does not search in RNN space.
>
> Since Bayesian optimization is similar to DARTS, the final architecture is selected by pruning the edges with a gamma metric smaller than a pre-defined threshold after the search is complete. One search will produce exactly one architecture. The average searched model performance on CIFAR-10 is 95.99 over 10 different searches, with the best model achieving 96.41. However, this performance is lower than the 96.48 achieved by our random baseline, as well as that of the other baseline search methods. With our training code, we reproduced the best architecture reported in BayseNAS (with a reported accuracy of 97.19), obtaining a similar performance of 97.13.
>
> With this additional baseline method, our conclusions remain unchanged. This further evidences our contribution of having a systematic way to evaluate NAS.
>
> Due to the limited rebuttal time period, we could not afford to perform more experiments. We will adapt BayesNAS to our reduced experiments in the future.

---

### Official Review · AnonReviewer1 · 2019-10-25
**Official Blind Review #1**

**Rating:** 6

**Review:**

This paper studies an important problem, evaluating the performance of existing neural architecture search algorithms against a random sampling algorithm fairly.

Neural architecture search usually involves two phases: model search and model tuning. In the search phase, best architectures after limited training are selected. In model tuning, the selected architectures are trained fully. However, it has been noticed that best architectures after limited training may not translate to globally best architectures. Although previous research has tried comparing to random sampling, such as Liu et al. 2019b, but the random architectures were not trained fully. The authors train random architectures fully before selecting the best one, which turns out to perform as well or better than the sophisticated neural architecture search methods. The paper also identifies that parameter sharing turns out to be a major reason why the sophisticated NAS methods do not really work well.

The insights are obviously important and valuable. The insight on parameter sharing is even a bit disheartening. Parameter sharing is the main reason why NAS can scale to very large domains. Without it, is NAS still practical or useful? On the other hand, it is a bit unsatisfactory that the paper does not provide or even suggest solutions to remedy the identified issues.

Another comment is it is a stretch to consider the evaluation done in the paper a new framework. It is simply a new baseline plus a new experiment design.

About Equation (1) in Appendix A.2, it seems to simplify to p=(r/r_max)^n. Is the formula correct?

**Experience Assessment:**

I have read many papers in this area.

**Review Assessment: Checking Correctness Of Derivations And Theory:**

I assessed the sensibility of the derivations and theory.

**Review Assessment: Checking Correctness Of Experiments:**

I assessed the sensibility of the experiments.

**Review Assessment: Thoroughness In Paper Reading:**

I read the paper at least twice and used my best judgement in assessing the paper.

---

> ### Author Response · Authors · 2019-11-11
> **Response to AnonReviewer1**
>
> Thank you for your time reading our paper and provide the review. We provide our response to your questions and our view of NAS in general.
>
> Practicality of NAS:
> Without weight sharing (WS) NAS is useful. With WS, the results are disappointing, but not hopeless. For instance, the kendall tau value of 0.2 we obtained in CNN space, i.e., not 0 as in RNN space, means that, with multiple runs, it is “possible” to find a better architecture. Our argument is rather that, with the WS, NAS still has a long way to go, and the community should study the lower-level behavior of the algorithms  instead of focusing on beating the state of the art with a point estimate of these algorithms.
>
> Framework:
> We refer to our study as a framework because we propose a systematic evaluation pipeline. One can use the same framework to compare different NAS algorithms in a fair way.
>
> Equation (1) in Appendix A.2 should be 1 - (1 - (r/r_max))^n, i.e., there was a typo in the outer bracket. Thanks for pointing it out.

---

### Official Review · AnonReviewer3 · 2019-10-28
**Official Blind Review #3**

**Rating:** 6

**Review:**

This works studies the evaluation of search strategies for neural architecture search. It points out existing problems of the current evaluation scheme: (1) only compares the final result without testing the robustness under different random seeds; (2) lacking fair comparison with random baseline under different random seeds. The authors analyzed three popular NAS methods with weight sharing (ENAS, DARTS, NAO), and showed that they don't significantly improve upon random baseline on PTB and CIFAR-10. On a reduced search space of RNN and CNN (NASBench), they showed that the three methods fail to find the best performing architecture. Then they compared search with and without weight sharing and showed the correlation between architecture performance under the two conditions in a reduced search space, which indicates the weight sharing is a potential cause for the suboptimal performance.

I recommend acceptance of the paper for the reasons below.

(1) It pointed out some important issues in the evaluation of NAS methods: evaluating under different random seeds and fair comparison with random baseline.
(2) The analysis is supported by experiments in the original search space and a reduced search space, which makes the result more convincing.
(3) It proposed the weight sharing as a potential cause and supported the hypothesis with experiments in the reduced search space, although more experiments in a realistic search space are needed to make the conclusion more solid.

Weakness:

(1) The problem that the search space is over-optimized and constrained is not unnoticed before. For example, table 1 in (Liu et al, 2018) showed that the random search baseline performs not much worse than the DARTS (~0.53% difference), which is similar to the conclusions on CIFAR-10 presented in this work.
(2) More recent works in NAS is already evaluating under multiple random seeds and performing fair comparison with random search baselines, for example, (So et al, 2019). There should be more discussions about such improvements in the rigorous evaluation of NAS.
(3) The comparison between with and without weight sharing in section 4.3 is interesting, but there should be more support in a realistic search space, because the landscape could be very different. Otherwise, it is better to make clear the scope of the conclusion, for example, instead of "in CNN space, the ranking disorder ...", it is better to use "in a reduced CNN space, ...".

"Darts: Differentiable architecture search." Liu, Hanxiao, Karen Simonyan, and Yiming Yang.  ICLR, 2019
"The Evolved Transformer." David R. So, Chen Liang, and Quoc V. Le., International Conference on Machine Learning. 2019.

Typos:
"based one their results on the downstream task." -> "based on"
"obtained an an accuracy" -> "obtained an accuracy"

====================================

I have read the author response and would keep the same rating. The paper pointed out an important issue, but it has also been noticed before. The insight on weight sharing is interesting, although more experiments are needed to testify the claim over state-of-the-art NAS search space.

**Experience Assessment:**

I have published one or two papers in this area.

**Review Assessment: Checking Correctness Of Derivations And Theory:**

N/A

**Review Assessment: Checking Correctness Of Experiments:**

I carefully checked the experiments.

**Review Assessment: Thoroughness In Paper Reading:**

I read the paper thoroughly.

---

> ### Author Response · Authors · 2019-11-11
> **Response to AnonReviewer3**
>
> Thank you for your detailed review and we provide our response to your questions one by one.
>
> (1) Liu et al, 2018, focus on the performance of the one best architecture found by their algorithm, without showing its average performance. By contrast, here, we argue that the community should study the lower-level behavior of the NAS algorithms, and in particular the searching policy. Furthermore, we identify the reasons for the disappointing average performance of NAS algorithms.
>
> (2) Thanks for pointing this work to us. We will discuss it in our related work section. Note that (So et al., 2019) perform NAS without weight sharing and thus require much larger computational resources than the algorithms we evaluated in our paper.
>
> (3) Our analysis was possible for NASBench, which already has 423K architectures, because NASBench gives access to architectures trained independently. However, the computational complexity to repeat this in a more realistic search space is equivalent to training millions of architectures from scratch, which is technically impossible given our resources. We nonetheless believe that our results on the half a million NASBench architectures already clearly support our claim that weight sharing in CNN space limits the NAS performances. We will revise the wording to clarify this.
>
> We will correct the typos as suggested. Thanks again for pointing them out.

---

### Author Response · Authors · 2019-11-15
**Updated manuscript and code**

We thank all reviewers for their time to read and comment on our paper. We update our manuscripts based on those comments.

- Adding NAS with Bayesian approach into discussion
- BayseNAS CNN space experiments as R2 suggested.
- Fixing Equation(1) in A.2 as R1 pointed out.
- Tuning down section 4.3 conclusion as R3 suggested.
- Fixing typos

The code for BayesNAS is at
https://drive.google.com/file/d/1d8KuUmmnorzia6FVKzNzBHIStqZSmefp/view?usp=sharing

---

### Decision · Program_Chairs · 2019-12-19

**Decision:**

Accept (Poster)

**Comment:**

This is one of several recent parallel papers that pointed out issues with neural architecture search (NAS). It shows that several NAS algorithms do not perform better than random search and finds that their weight sharing mechanism leads to low correlations of the search performance and final evaluation performance. Code is available to ensure reproducibility of the work.

After the discussion period, all reviewers are mildly in favour of accepting the paper.

My recommendation is therefore to accept the paper. The paper's results may in part appear to be old news by now, but they were not when the paper first appeared on arXiv (in parallel to Li & Talwalkar, so similarities to that work should not be held against this paper).